# Tribological Performance of MoS_2_–WS_2_ Composite Film under the Atomic Oxygen Irradiation Conditions

**DOI:** 10.3390/ma13061407

**Published:** 2020-03-19

**Authors:** Yanlong Fu, Dong Jiang, Desheng Wang, Xiaoming Gao, Ming Hu, Wu Yang

**Affiliations:** 1College of Chemistry and Chemical Engineering, Northwest Normal University, Lanzhou 730070, China; fyanl@licp.cas.cn; 2State Key Laboratory of Solid Lubrication, Lanzhou Institute of Chemical Physics, Chinese Academy of Sciences, Lanzhou 730000, China; wangdsh@licp.cas.cn (D.W.); gaoxm@licp.cas.cn (X.G.); hum413@licp.cas.cn (M.H.)

**Keywords:** MoS_2_–WS_2_ composite film, atomic oxygen (AO), tribological properties

## Abstract

The oxidation and erosion of atomic oxygen are considered to be the most dangerous environmental factors for materials exposed to the aerospace environment. In order to investigate the effect of atomic oxygen on the lubricating film and improve the tribological properties, MoS_2_–WS_2_ composite film was prepared by the sputtering technique. The film structure and mechanical behavior were characterized and their vacuum tribological properties were evaluated by a friction tester. The composite film possessed better atomic oxygen (AO) resistance than pure film because of the dense structure. The tribological performance of composite film was different under the conditions after AO-irradiation and sliding and during AO-irradiation. After AO-irradiation, the tribological properties of composite film were similar to those before AO-irradiation. However, high friction noise, wear rate, and reduced wear duration were observed for the composite film under the AO-irradiation/friction process because of the continuous damage of the lubricating film due to the AO-irradiation. The addition of 16 at.% WS_2_ to the MoS_2_-based film changed the composite film structure and improved the oxidation resistance of the film, making the composite film exhibit better tribological performance than pure MoS_2_.

## 1. Introduction

It is well known that the orbits of most spacecrafts are about 100–1000 km from the ground, which are the low-earth orbits (LEO) [1,2,3,4,5]. In these orbits, the oxidation and erosion of atomic oxygen are considered to be the most dangerous environmental factor for materials exposed to the aerospace environment [6,7,8,9,10]. Therefore, more attention has been given in recent years to study the interaction between atom oxygen and space materials including polymers and solid lubricants.

Molybdenum disulfide (MoS_2_), as a typical solid lubricating film, is widely used in aerospace fields due to its better mechanical properties and excellent tribological properties in vacuum and inert gas environments [11,12,13,14]. However, its wear resistance is affected by the porous columnar microstructure of the film. In order to solve the problem, many researchers improved the mechanical and tribological properties of the film by compounding and multi-layering. Simmonds et al. [15,16] found that the addition of 10% WSe_2_ could promote the crystallization of the MoS_2_ film and the growth of the (002) crystal plane orientation, thereby improving the friction and wear properties of the film in the atmosphere. Watanabe et al. [17] found that the MoS_2_–WS_2_ multilayer film exhibited lower friction coefficient and longer service life than the pure MoS_2_ and WS_2_ films. Guilin et al. [18] prepared the MoS_2_–WS_2_ composite lubricating film by magnetron sputtering and studied the tribological properties in the atmosphere. The results showed that the addition of WS_2_ did not promote the preferential growth of the (002) crystal plane. However, the composite film possessed a dense structure and exhibited superior tribological properties than the pure MoS_2_ film. Jian et al. [19] found that the friction properties of MoS_2_–WS_2_ composite films in dry air were better than those in humid air. Based on previous studies, the wear resistance and oxidation resistance of composite films containing WS_2_ were improved in the atmosphere. Nevertheless, atomic oxygen (AO) possesses higher chemical activity and kinetic energy than molecular oxygen (MO), and its oxidation behavior could be different. For example, silver could not be oxidized by MO at temperatures below 350 °C and oxygen pressures below 1.3 × 10^3^ Pa but was severely oxidized by AO [20,21,22].

In this paper, MoS_2_–WS_2_ composite film was prepared by radio-frequency (RF) magnetron sputtering technique, and the surface and cross-section morphologies, crystal structure, and thickness of the films were characterized. The mechanical properties of MoS_2_–WS_2_ composite films were investigated, and the different tribological mechanisms of composite film were explored under single atomic oxygen irradiation and simultaneous atomic-oxygen/friction conditions to distinguish their failure behavior.

## 2. Materials and Methods

### 2.1. Preparation of MoS2–WS2 Composite Films

The MoS_2_–WS_2_ composite film was deposited by Ø80 mm hybrid target through a radio-frequency magnetron sputtering system. The structure of the sputtering device is shown in Figure 1.

The composite target was composed of MoS_2_ and WS_2_, the purities of which were 99.9%. The powders of MoS_2_ and WS_2_ were mixed and pressed into one hybrid target, the mass ratio of which was 1:2. The substrates used in the experiment were an AISI 440C steel sheet and an Si sheet. The steel sheets were used to test the mechanical properties and tribological properties of the composite film. The Si sheets were used to characterize the surface and cross-section topography and structure of the film. Before deposition, the surfaces of the steel substrates were sanded and polished to Ra ≤ 0.03 μm, ultrasonically cleaned in acetone and dried, then fixed on the sample holder in the vacuum chamber. The vacuum chamber was pumped to a low vacuum of 1.0 × 10^−3^ Pa. Additionally, Ar gas was used as the working gas source. Before the film was deposited, in order to eliminate possible contaminant on the surface of the substrate, the surface of the substrate was washed with Ar ions at a bias of −500 V for 15 min. Thereafter, composite films were deposited at 500 W of sputtering power and 4.0 Pa of gas pressure. The deposition time was 20 min, and the film thickness was determined to be about 2 μm. For comparison, pure MoS_2_ film was deposited under the same conditions. The specific process parameters are shown in Table 1.

### 2.2. Structure and Properties of the Films

The surface and cross-section morphologies were examined with a JS-6701 scanning electron microscope (FESEM) attached with an X-ray energy-dispersive spectroscopy (EDS). The crystal structure was measured with X-ray diffraction (XRD) (Bruker, Germany) using Cu Ka radiation with a scanning rate of 0.02 °/s. The film hardness and elastic modulus were measured using a nano indenter (TI-950, Hysitron, Minneapolis, MN, USA). During the test, the diamond indenter was loaded and unloaded linearly. To avoid the influence of the substrate in the hardness test, the indentation depth was 100 nm, which did not exceed 10% of the film thickness. Each film sample was tested at least three times, and the arithmetic average of the test data was taken as the hardness value and elastic modulus value of the film.

The adhesion strength of composite film was measured with a scratch tester (Kaihua MFT-4000), with a conical diamond tip of 0.2 mm radius and 120° taper angle. The test conditions were as follows: a scratch length of 5 mm, termination load of 100 N, and loading rate of 100 N/min. The friction force and friction coefficient were recorded in the scratch tests. After the scratch test, the scratched surface of the sample was observed through an optical microscope. The adhesion strength value was finally expressed by the critical load (Lc) of the film peeling off from the surface of the substrate.

AO-irradiation tests on the lubricating films were implemented via a ground simulation device, in which atomic oxygen was produced. Under an electromagnetic field, the oxygen plasma produced by the microwave power source was accelerated towards a molybdenum plate and was neutralized by negative charges and rebounded to form a neutral AO beam with an average impingement kinetic energy of ~5 eV. The AO flux was determined to be 8.4 × 10^19^ atoms/(cm^2^·s). The chemical compositions of the film before and after atomic oxygen irradiation were characterized by X-ray photoelectron spectroscopy (XPS, PHI) with an argon sputtering gun equipped with monochromatic Al Ka rays. The sputtering rate was 12.5 nm/min.

Their tribological performances were evaluated by a ball-on-desk tribo-tester under a vacuum environment (≤1.3 × 10^−3^ Pa). The schematic representation of tribo-tester is shown in Figure 2. The friction sensor model of the data acquisition system is C3S2U, which is produced by China Metrology Technology Development Corporation. The sensor precision is 0.005, and the acquisition rate is 5 data per minute. The counterpart was an AISI 440C steel ball of 8 mm in diameter. The conditions were as follows: three kinds of sliding rates, namely 20 r/min, 100 r/min, and 200 r/min; a normal load of 5 N; and a test duration of 60 min. The friction coefficient was recorded automatically by computer.

## 3. Results and Discussions

### 3.1. Surface and Cross-Section Morphologies of the Deposited Film

The surface and cross-section morphologies of the as-obtained films are shown in Figure 3. The thicknesses of composite film and pure MoS_2_ films were 1.72 μm and 1.83 μm, respectively. The addition of 16 at.% WS_2_ was doped into the composition film, which was measured with EDS. Similar to the pure MoS_2_ film, the typical dendritic structure feature was presented on the composite film surface. Compared to the thick columnar crystals and high porosity of pure MoS_2_ film, the composite film possessed lower porosity, which was shown in the cross-section morphology. It was attributed to the effect of doping WS_2_ and the crystal anisotropy on the film structure.

The XRD diffraction patterns of the MoS_2_–WS_2_ composite film and pure MoS_2_ film are shown in Figure 4. There were (100), (101), and (112) diffraction peaks besides the weak (002) diffraction peak for the two films. Since WS_2_ and MoS_2_ both belong to the transition metal disulfides and have similar lattice constants, the MoS_2_–WS_2_ composite film and pure MoS_2_ film had similar diffraction peaks. Based on previous experiments, the preferred orientation of crystal face in thin films can be changed by doping, optical deposition parameters, multi-layer construction, and so on. Simmonds et al. [15,16] found that the addition of 10% WSe_2_ to MoS_2_ films could promote the crystallization of the film and the (002) crystal plane orientation growth. Doping of WS_2_ would promote the preferred growth of the composite film with the (002) basal plane, making the film denser, with reduced porosity and higher hardness of the film.

Surface energy–strain energy theory is usually used to explain the preferential orientation growth of thin film crystal planes. In the theory, due to the small stress, the crystal surface with a small surface energy possesses some certain advantage and shows preferential growth during the early stage of film growth. Thereafter, the film stress gradually increases as the film thickens, and the crystal face with lower strain energy presents preferential growth.

Figure 5a shows the histogram of hardness (H), elasticity modulus (E), H^3^/E^2^, and elastic recovery of the as-obtained pure MoS_2_ film and MoS_2_–WS_2_ composite film. The loading and unloading curves of the films are shown in Figure 5b. It could be seen that the hardness of pure MoS_2_ film was about 0.25 GPa, which was basically consistent with the hardness value of MoS_2_ films with porous columnar crystal structure in the literature [23,24]. The hardness of the composite film was about 1.67 GPa. The elastic recovery rate of MoS_2_–WS_2_ composite film was about 35%, which was higher than that of MoS_2_ film (about 20%). In addition, the H^3^/E^2^ value of MoS_2_–WS_2_ composite film was also much higher than that of pure MoS_2_ film. Some studies have confirmed that doping metals and compounds can ameliorate the film crystal structure and increase the film hardness, which is mainly due to the densification of the film structure and the solid solution strengthening effect [25]. The as-obtained MoS_2_–WS_2_ composite film had a dense structure, which was the most important reason for the high hardness of the composite film.

The film-base adhesion strength of the composite film on the stainless steel substrate was further studied by the scratch method. Figure 6 shows the scratch curves of MoS_2_–WS_2_ composite film and pure MoS_2_ film and the optical micrographs after the scratch tests. The adhesion strength (Lc) of the pure MoS_2_ film on the steel substrate was 35.46 N, and the adhesion strength (Lc) of the MoS_2_–WS_2_ composite film was 47.10 N, as shown in Figure 6b. Compared with pure MoS_2_ film, the composite film possessed stronger adhesion strength on the substrate. Shown in the scratch micrographs, the scratch edge of the MoS_2_–WS_2_ composite film was relatively smooth, and the film debris was fine-particle; especially, there was no large peeling phenomenon. Previous studies have proved that the doping and compound technology can improve the diffusion and chemical bonding of deposited particles on the film-based interface, thereby increase the film-based adhesion strength [26,27]. The addition of WS_2_ could promote interfacial bonding and make the composite film possess higher hardness. Based on the above tests, the composite film exhibited better toughness and bonding performance with stainless steel substrate.

In order to further analyze the atomic oxygen resistance of the composite film, the chemical states of Mo and W elements at different depth of the MoS_2_–WS_2_ composite film after atomic oxygen irradiation were explored. The analysis results are shown in Figure 7. For W chemical element (Figure 7a), WO_3_ and WS_x_O_y_ were detected on the top-surface of the composite film. With the increase of the etching time, the peaks of W4f_7/2_ and W4f_5/2_ shifted in the direction of lower binding energy, and the intensity of the WO_3_ peak continued to decrease. After 2 etching minutes, the peak of WO_3_ spectrum basically disappeared. When the etching time was 3 minutes, WO_2_, WS_x_O_y_, and WS_2_ were detected in the film. When the etching time was 5 min, WS_x_O_y_ and WS_2_ were only detected in the film. Thereafter, the shape and position of the spectrum of W4f remained basically unchanged. This indicated that the atomic oxygen irradiation mainly occurred on the surface of the composite film, and the W element mainly existed in the form of WS_x_O_y_ and WS_2_ in the film. Figure 7b shows the depth spectrum of Mo 3d in the composite film. MoO_2_ and MoS_x_O_y_ were also detected on the top-surface of the composite film. As the etching time progressed, the MoO_2_ peak basically disappeared, and the Mo 3d_5/2_ and 3d_3/2_ peaks were almost unchanged after 5 etching minutes. Within the film, the Mo 3d peaks were mainly attributive to MoS_x_O_y_ and MoS_2_ peaks. Pure MoS_2_ film was easily oxidized due to the porous structure under the AO-irradiation condition, and the oxidation depth was 125 nm. The oxidation depth of composite film reduced to half of that for pure film.

During the sputtering deposition process, the residual O atoms from the vacuum chamber and the target surface replaced the S atoms in the MS_2_ lattice, which led to the partial oxidation of the film to MS_x_O_y_ (M is W and Mo). After atomic oxygen irradiation, M atoms in the film were oxidized to MO_3_, MO_2_, and MS_x_O_y_, so the O content increased. Additionally, some S atoms were lost due to oxidation to form sulfur oxides. Studies have shown that the degree of oxidation of WS_2_ and MoS_2_ crystals by atomic oxygen depends on their crystal plane orientation. The basal plane oriented crystals have high resistance to atomic oxygen [28,29]. The MoS_2_–WS_2_ composite film prepared in this experiment had strong (002) preferred orientation, making the composite film possess stronger atomic oxygen resistance. In addition, the denser structure of composite film than pure MoS_2_ film also prevented the diffusion and penetration of atomic oxygen into the interior of the film effectively.

### 3.2. Tribological Performance of Composite Films Under AO Environment

Under the LEO environment, atomic oxygen has a significant impact on the structure and performance of space lubricating films. Due to the limitations of space experimental conditions, it is difficult to carry out atomic oxygen irradiation experiments in real aerospace environment. In order to investigate the effect of atomic oxygen on the structure and tribological properties of the thin films, tribological tests were carried out in the following two states: (1) tribological tests were performed after AO-irradiation, and (2) tribological tests were performed simultaneously with AO-irradiation. The atomic oxygen flux was 8.4 × 10^19^ atoms/(cm^2^∙s)).

#### 3.2.1. Tribological Performance of Composite Films after AO-Irradiation 

After 60 min atomic oxygen irradiation test, the tribological properties of two films (pure MoS_2_ film and MoS_2_–WS_2_ composite film) were tested by a ball–disk friction tester. The real-time vacuum friction coefficient curves of the two films at different speeds are shown in Figure 8, and the corresponding wear rates are listed in Figure 10. When the rotation speed was 20 r/min (Figure 8a), the average friction coefficient of the MoS_2_ film was 0.058 before irradiation and 0.062 after irradiation, and the average friction coefficient of the composite film was 0.046 before irradiation and 0.048 after irradiation. When the rotation speed was 100 r/min (Figure 8b), the average friction coefficient of MoS_2_ film was 0.040 before irradiation and 0.048 after irradiation, and the average friction coefficient of the composite film was 0.035 before irradiation and 0.042 after irradiation. At 200 r/min (Figure 8c), the average friction coefficient of the MoS_2_ film was 0.040 before irradiation and 0.045 after irradiation, and the average friction coefficient of the composite film was 0.035 before irradiation and 0.038 after irradiation. It could be seen that the friction coefficient and wear rate (Figure 10) for each film under the same friction conditions had not changed significantly, and they both demonstrated good lubrication performance; but after atomic oxygen irradiation, the friction noise of the two films were larger than those before the atomic oxygen irradiation.

At different speeds, the friction coefficient and wear rate of the composite film were lower than those of pure MoS_2_ films, and as the speed increased, the friction coefficient and wear rate gradually decreased. The friction coefficient and the wear rate were lowest at the sliding speed of 200 r/min. It was because that the oxidation phenomenon only occurred on the top-surface of the film after the atomic oxygen irradiation test. After the initial sliding stage, the oxide film of the film on the surface was worn away, and thus the fresh lubricating film was exposed. Afterwards, friction occurred between the steel ball and the fresh lubricating film, and a transfer film was formed between the counterparts. So their friction coefficient decreased and stabilized. At the same time, the (002) crystal plane orientation of the film had a significant effect on the friction coefficient, which was more conducive to improve lubrication performance [11,14]. Because the composite film contained higher content of the (002) orientated crystal plane than pure MoS_2_, the composite film exhibited better tribological properties.

#### 3.2.2. Tribological Properties of Thin Films During Atomic Oxygen Irradiation

Figure 9 shows the real-time friction curves of MoS_2_ film and MoS_2_–WS_2_ composite film under the simultaneous action of AO-irradiation/friction at different sliding speeds. At lower speed of 20 r/min, both films exhibited low friction coefficients and high wear duration. When the rotation speed was 20 r/min (Figure 9a), the average friction coefficient of the MoS_2_ film was 0.045 and the average friction coefficient of the composite film was 0.042. When the rotation speed was 100 r/min (Figure 9b), the average friction coefficients of MoS_2_ film and the composite film were 0.043 and 0.054, respectively. At 200 r/min (Figure 9c), the average friction coefficients of MoS_2_ film and the composite film were 0.042 and 0.058, respectively. Moreover, the friction coefficients of pure MoS_2_ increased as the sliding time prolonged at the rotation speed of 100 r/min and 200 r/min. No significant change of the friction coefficient was observed for composite MoS_2_–WS_2_ film. At higher speeds of 100r/min and 200r/min, the maximum instantaneous friction coefficients of pure MoS_2_ film were higher than 0.15 when the number of rotations was 4.5 × 10^3^ cycles and 6.6 × 10^3^ cycles, respectively. This meant that the lubricating film was failing. However, the MoS_2_–WS_2_ composite film exhibited better lubrication performance, showing lower friction coefficient and longer wear duration than pure MoS_2_ film. The wear rates of the two films were calculated when the number of rotations was 1.2 × 10^3^ cycles, 6.0 × 10^3^ cycles, and 1.2 × 10^4^ cycles, as shown in Figure 10. The wear rates of MoS_2_ film and the composite film were 11.24 × 10^−5^ mm^3^∙N^−1^∙m^−1^ and 5.27 × 10^−5^ mm^3^∙N^−1^∙m^−1^ at 20 r/min, 8.59 × 10^−5^ mm^3^∙N^−1^∙m^−1^ and 2.75 × 10^−5^ mm^3^∙N^−1^∙m^−1^ at 100 r/min, and 2.49 × 10^−5^ mm^3^∙N^−1^∙m^−1^ and 1.75 × 10^−5^ mm^3^∙N^−1^∙m^−1^ at 200 r/min, respectively. Compared with pure MoS_2_ film, the wear rates of composite film were reduced 53% at 20 r/min, 68% at 100 r/min and 30% at 200 r/min. It could be seen that the wear rates of the two films were different at different speeds. Although the wear rates after atomic oxygen irradiation were high, the wear rate of the composite film was lower than that of pure MoS_2_ film, which was 1/2 of that of pure MoS_2_ film. This was mainly because the addition of WS_2_ changed the film structure and improved the oxidation resistance and the tribological properties of MoS_2_–WS_2_ composite films significantly.

After atomic oxygen irradiation, the MoS_2_-based solid lubricating film exhibited similar tribological properties to those before AO-irradiation under vacuum conditions. However, the changes of their tribological properties were more obvious under the simultaneous AO-irradiation/friction process, which not only contained high friction noise and wear rate, but also had a significant reduction in wear duration. The severe abrasion of the solid lubricating film was related to the oxide formed by continuous atomic oxygen irradiation and the peeling degree of oxide during the friction process. Previous studies indicated that atomic oxygen damage of the lubrication film could be alleviated, and the abrasion resistance could be improved by increasing the density of the film [30]. However, in the atomic oxygen irradiation–friction process, there were two aspects to solve, namely the denseness and the oxidation resistance of MoS_2_-based films, which finally improved the tribological performance of the solid lubricating films in the real space exposed environment.

### 3.3. Analysis of Friction and Wear Mechanism

In order to further study the friction and wear mechanism of lubricating films under atomic oxygen irradiation conditions, the morphology of wear tracks of the two films at different speeds and the same test time was characterized (Figure 11 and Figure 12). It could be seen that their wear tracks were narrow, and their surfaces were smooth after atomic oxygen irradiation (Figure 12a,b). Because the oxidative damage only occurred on the film surface, the oxide on the film surface was worn away easily in a short rubbing time, and the fresh lubricating film was exposed. Since then, friction would be taken place between the steel ball and fresh lubricating film. So, their tribological properties after AO-irradiation were similar to those before AO-irradiation.

Obvious flake wear debris and severe abrasive wear could be observed in the wear tracks of the two solid lubricant films under the simultaneous AO-irradiation/friction process. It was mainly because the oxide film on the film surface was worn out, and the fresh film surface exposed continued to oxide under the simultaneous AO-irradiation/friction conditions, as shown in Figure 11a,b. So, the friction mainly occurred between the oxide and counterpart. Because the oxide did not possess good lubricity, the lubricating transfer film could not be formed in the friction area. However, the wear track of the composite film was smoother with fewer wear flakes than that of pure MoS_2_. During the AO-irradiation/friction process, the flake debris on the wear track peeled off from the film induced by the fatigue, and AO-irradiation was observed at low speed (20 r/min), and the granular wear debris on the wear track was found at a higher speed (200 r/min). When the rotation speed was low, the irradiation time of the atomic oxygen to the film was relatively long, and continuous oxidation action was carried out on the film surface. The wear mechanisms were mainly oxidative wear, abrasive wear, and fatigue wear. With the increase of the rotation speed, the wear mechanisms were mainly fatigue wear and abrasive wear. The addition of WS_2_ to the MoS_2_-based film changed the film structure and improved the oxidation resistance of the film, making the composite film exhibit better tribological performance than pure MoS_2_.

## 4. Conclusions

The as-obtained composite film possessed a dense film microstructure and exhibited good atomic oxygen resistance because of the doping of WS_2_. Compared with the pure MoS_2_ film, there was no significant change of the surface morphology for the composite film after atomic oxygen irradiation under the atomic oxygen radiation flux of 8.4 × 10^19^ atoms/(cm^2^·s). The oxidation depth of composite film decreased to half that of pure film. After the atomic oxygen irradiation, the tribological properties of the two films were similar to the lubricating properties before irradiation under vacuum conditions. This was because the oxide film of the film on the surface was worn away at the initial sliding stage; thus, the fresh lubricating film was exposed to ensure good tribological performance. However, during the simultaneous AO-irradiation/friction process, the composite film exhibited more excellent tribological properties than pure MoS_2_ film. The wear mechanisms were mainly oxidative wear, abrasive wear, and fatigue wear. The addition of WS_2_ to the MoS_2_-based film changed the film structure and improved the oxidation resistance of the film. Compared with pure MoS_2_ film, the wear rate of composite film was reduced 53% at 20 r/min, 68% at 100 r/min, and 30% at 200 r/min, making the composite film exhibit better tribological performance than pure MoS_2_.

## Figures and Tables

**Figure 1 materials-13-01407-f001:**
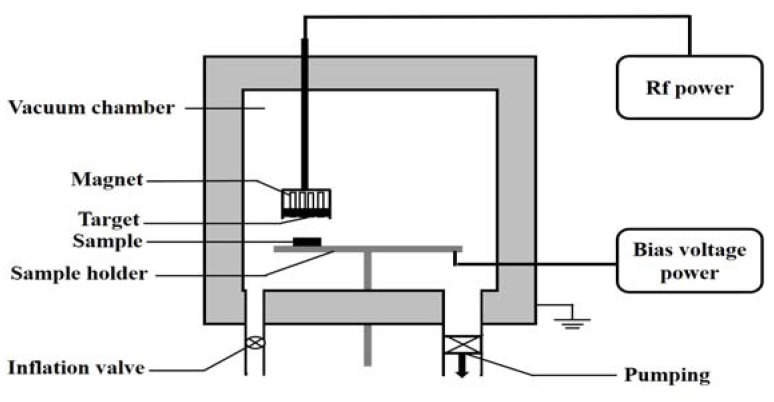
Schematic representation of radio frequency magnetron sputtering system.

**Figure 2 materials-13-01407-f002:**
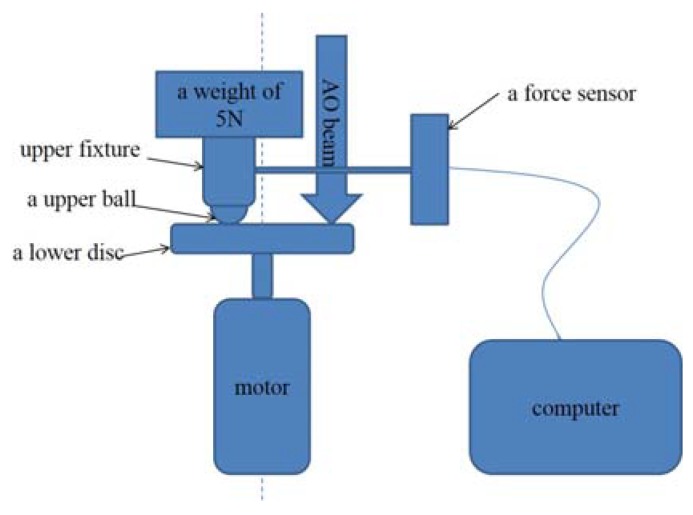
Schematic representation of tribo-tester.

**Figure 3 materials-13-01407-f003:**
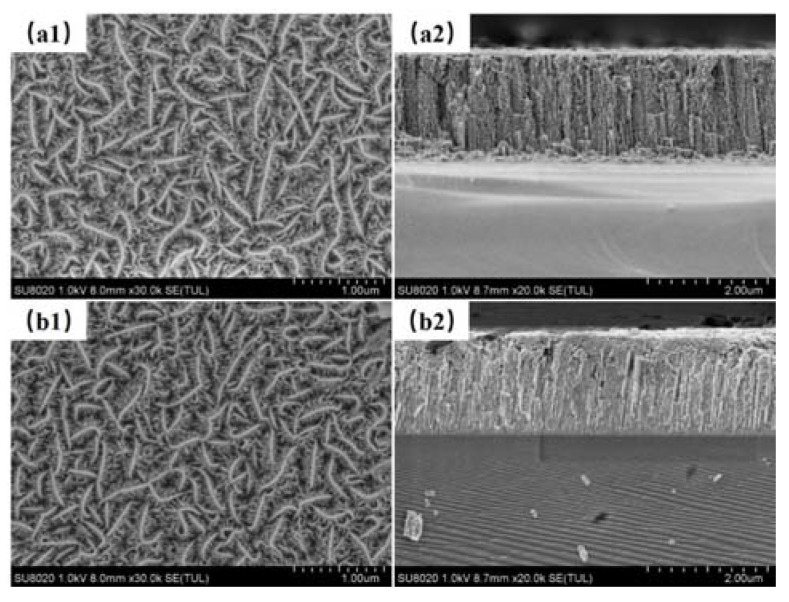
Surface and cross-section micrographs of pure MoS_2_ (**a1**,**a2**); MoS_2_–WS_2_ composite film (**b1**,**b2**).

**Figure 4 materials-13-01407-f004:**
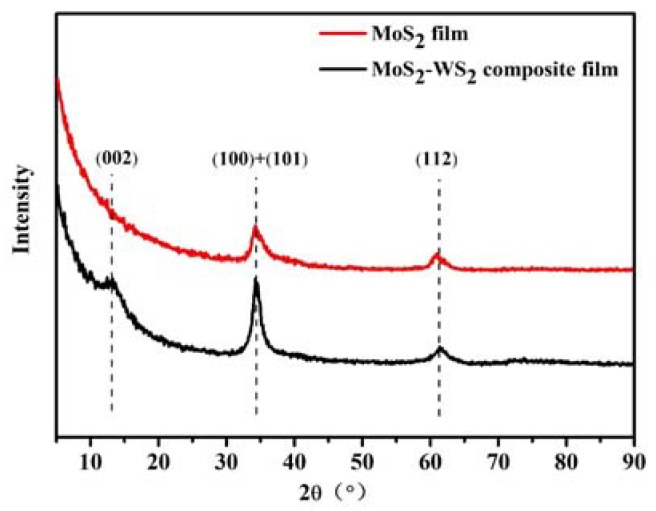
XRD diffraction patterns of pure MoS_2_ and MoS_2_–WS_2_ composite film.

**Figure 5 materials-13-01407-f005:**
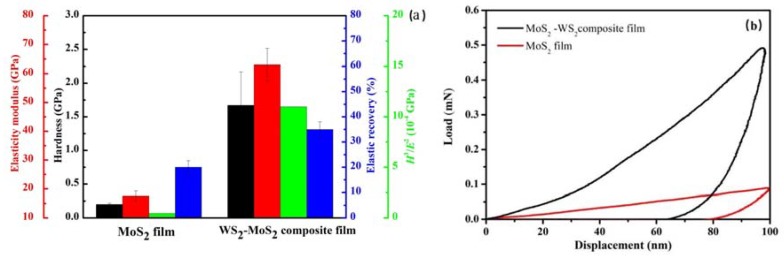
Hardness (*H*), elasticity modulus (*E*), H^3^/E^2^, and elastic recovery for pure MoS_2_ film and MoS_2_–WS_2_ composite film (**a**) and the typical load-unload curve (**b**).

**Figure 6 materials-13-01407-f006:**
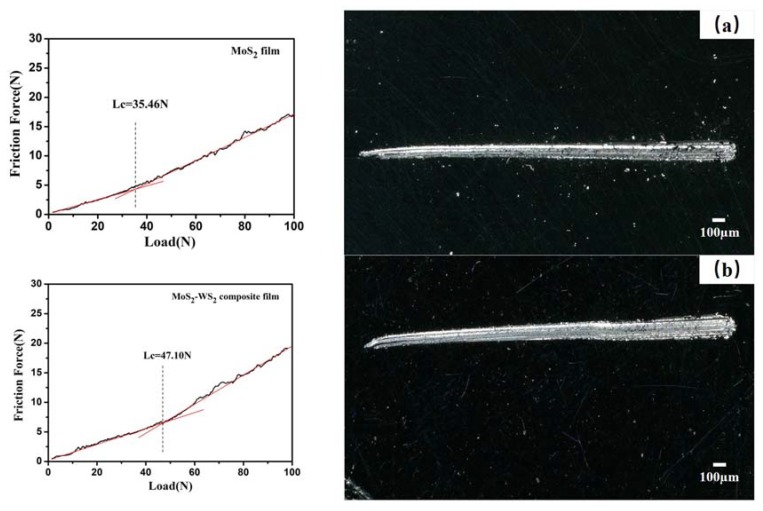
Scratch curves and tracks of MoS_2_ film (**a**) and MoS2–WS2 film (**b**).

**Figure 7 materials-13-01407-f007:**
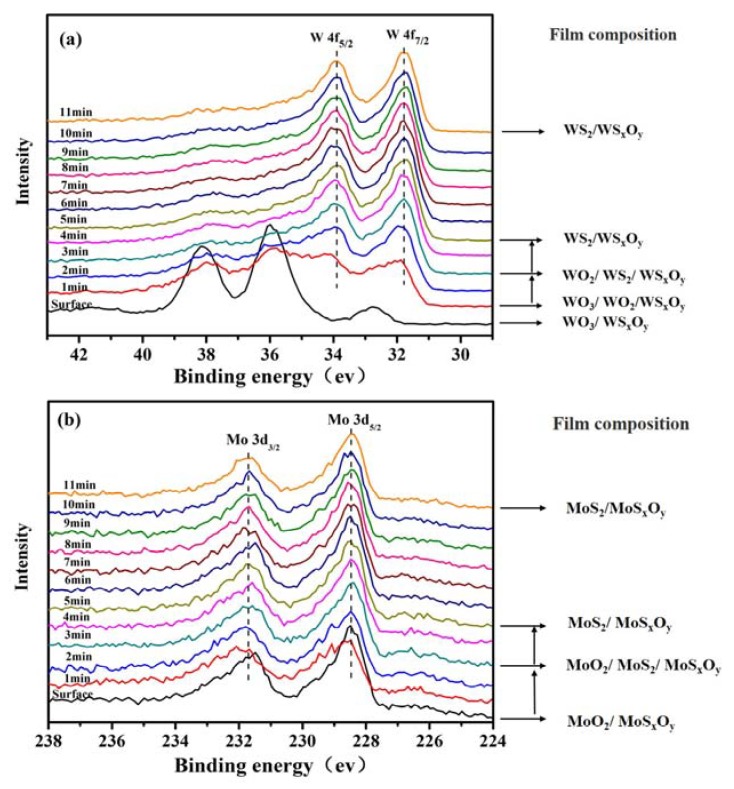
XPS depth profiles of MoS_2_–WS_2_ composite film: (**a**) W 4f, (**b**) Mo 3d after atomic oxygen irradiation.

**Figure 8 materials-13-01407-f008:**
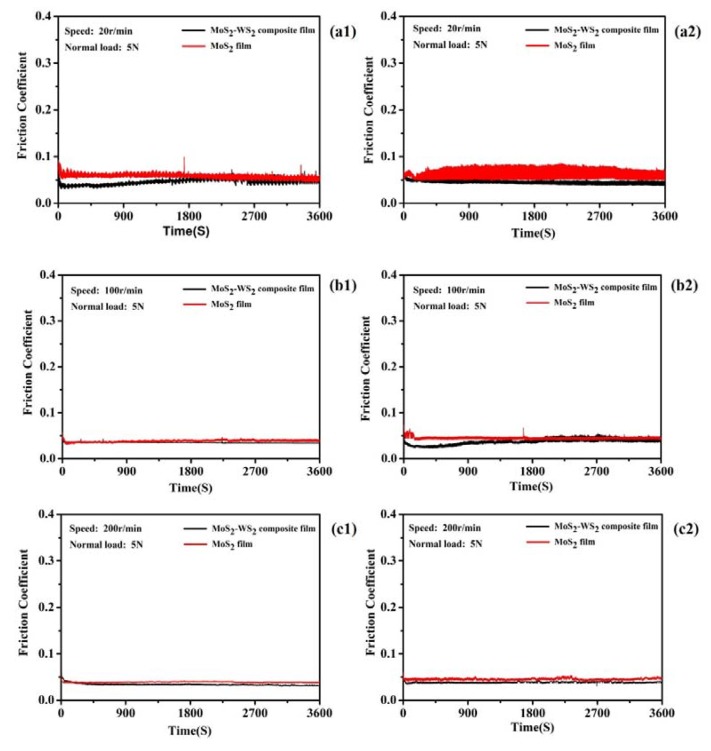
The real-time friction coefficient of pure MoS_2_ and MoS_2_–WS_2_ composite film before and after AO-irradiation under vacuum environment. (**a1**,**b1**,**c1**): before AO-irradiation; (**a2**,**b2**,**c2**): after AO-irradiation.

**Figure 9 materials-13-01407-f009:**
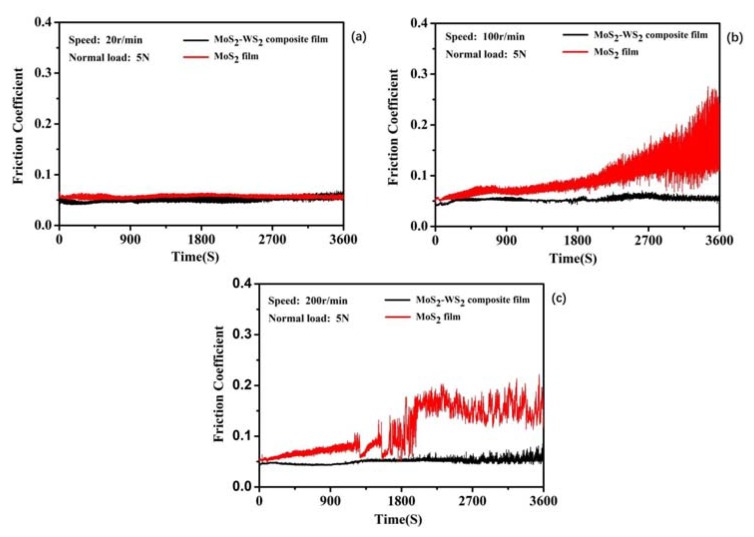
The real-time friction coefficient of pure MoS_2_ and MoS_2_–WS_2_ composite film at different sliding speed under AO-irradiation condition. (**a**) 20 r/min, 5N, (**b**) 100 r/min, 5N, (**c**) 200 r/min, 5N.

**Figure 10 materials-13-01407-f010:**
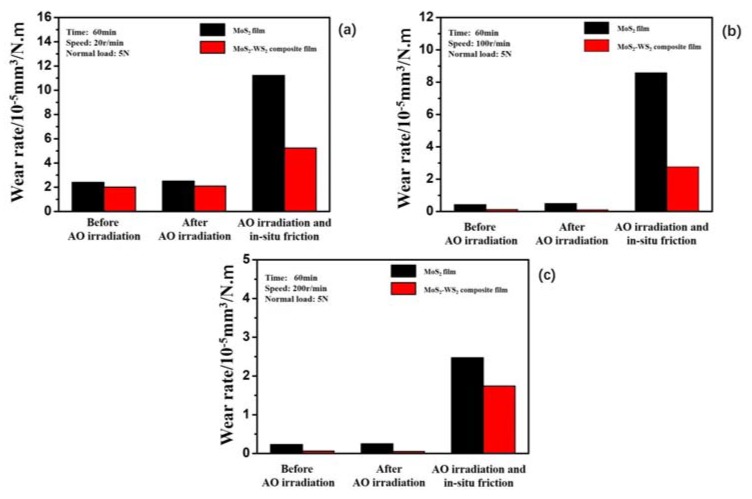
The wear rate of pure MoS_2_ and MoS_2_–WS_2_ composite film at different sliding cycles under different conditions: before AO-irradiation, after AO-irradiation, and with AO-irradiation. (**a**) 1.2 × 10^3^ cycles, (**b**) 6.0 × 10^3^ cycles, (**c**) 1.2 × 10^4^ cycles.

**Figure 11 materials-13-01407-f011:**
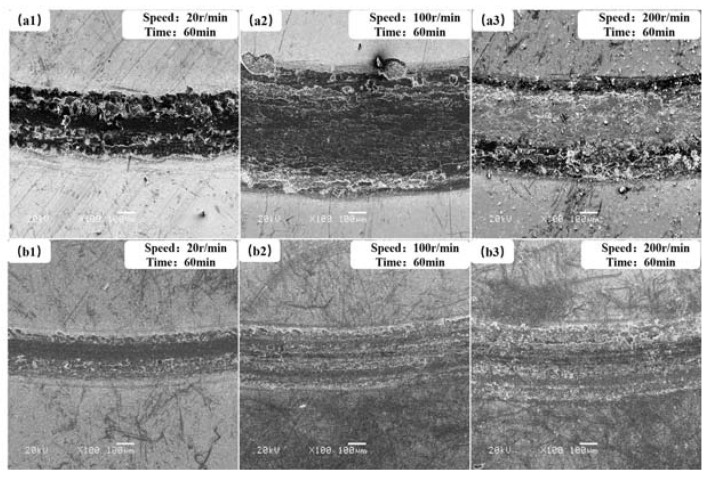
The morphologies of wear tracks for pure MoS_2_ film (**a1**–**a3**) and MoS_2_–WS_2_ composite film (**b1**–**b3**) under the vacuum and AO-irradiation at different sliding speeds.

**Figure 12 materials-13-01407-f012:**
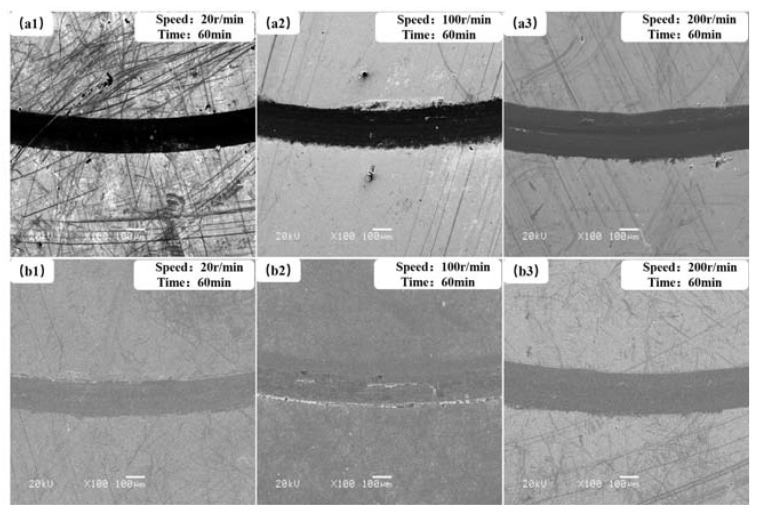
The morphologies of wear tracks for pure MoS_2_ film (**a1**)–(**a3**) and MoS_2_–WS_2_ composite film (**b1**)–(**b3**) after AO-irradiation at different sliding speeds.

**Table 1 materials-13-01407-t001:** The deposition parameters of composite and pure lubricating films.

Film Type	Process Parameters				
	Base Pressure (Pa)	Sputtering Power (W)	Bias Voltage (V)	Working Pressure (Pa)	Deposition Duration (min)
MoS_2_ film	≤1.3 × 10^−3^	500	−60	4.0	20 min
MoS_2_–WS_2_ film	≤1.3 × 10^−3^	500	−60	4.0	20 min

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
