# Peer review of "Tribological Performance of MoS2–WS2 Composite Film under the Atomic Oxygen Irradiation Conditions"

_materials, 2020, doi:10.3390/ma13061407_

Round 1
Reviewer 1 Report
The paper addresses a new and appealing research topic, the approach developed is quite clear. In order to enhance the quality of this paper, the following issues have to be addressed before considering the paper for the publication:
- The authors should explain (shortly) why the magnetron sputtering technique has been selected to obtain the composite film. I would also recommend adding some sentences with a short characterization of this technique to the Introduction or Materials and Methods part.
- In line 63, the authors wrote that “The composite target was composed of MoS2 and WS2, which purities were 99.9%”. In which form the composite target has been used? What was the ratio of MoS2 to WS2?
- I do not quite understand why Authors rely on references [15] and [16] concerned with the addition of WSe2 (lines 37-39 and 124-126), while the current investigation is concerned with WS2. If these compounds affect the properties of the proposed composite in a similar way it should be highlighted in this paper.
- In lines 126-128, the authors wrote that “Doping of WS2 would promote the preferred growth of the composite film with (002) basal plane, making the film denser, reduced porosity and higher hardness of the film”. This sentence overstates the results, especially in terms of (002) growth. The authors should add some references or expand the discussion on that subject. Additionally, in the Introduction part (lines 41-44) there is written that “Guilin et al [18] prepared the MoS2-WS2 composite (…) The results showed that the addition of WS2 did not promote the preferential growth of the (002) crystal plane”. If the current results show differently (e.g. presented XRD diffraction patterns), the authors should discuss this issue.
- Why in Figure 4, the standard deviation is presented only for hardness?
- Please consider adding scale instead of magnification value in Figure 5.
- There is some serious issue with the numeration of the figures in this version of the manuscript. On page 11 there are two figures no.10 and two figures no.11.
- Please add a reference to the sentence “Previous studies indicated that atomic oxygen damage of the lubrication film could be alleviated and the abrasion resistance could be improve by increasing the density of the film” (lines 252-254).
Minor issues (lines):
62.: Figure 1 is signed as “radio frequency sputtering system” while in the text there is “magnetron sputtering system”.
76, 77, 229, 259, 263, all descriptions on page 11.: Please, start the description with a capital letter.
Page 6 and description of Figure 6.: In some places you write “Mo3d” (line 173) and in other “Mo 3d” (line 176). Please unify.
197.: Delete one bracket at the end of the sentence.
264.: Add space between “(a)” and “1.2”
293.: Add space between “of” and “WS2”.
Reviewer 2 Report
In aerospace applications, liquid lubricants are not able to assure good lubrication conditions, because of extreme temperatures. In such conditions, solid lubricants are preferred, MoS2 and graphite based composite films being the main candidates. Even such performant solid lubricants are exposed to the highly oxidative environment, the action of atomic oxygen conducting to oxidation of the superficial layers. The paper investigates the tribological performances of MoS2-WS2 composite film, prepared by magnetron sputtering technique, in atomic oxygen (AO) irradiation and friction conditions.
Pin-on-disk tests are carried out in vacuum, the structure of MoS2 and MoS2-WS2 composite films being analyzed before and after the tests by optical microscopy and scanning electron microscopy (FESEM) attached with X-ray energy-dispersive spectroscopy (EDS). The crystal structure was measured with X-ray diffraction (XRD), and the film hardness and elastic modulus were measured using a nano indenter. Also, the adhesion strength of composite film was assessed by scratch tests, the chemical composition of film before and after atomic oxygen irradiation being characterized by X-ray photoelectron spectroscopy (XPS, PHI).
I appreciate the punctilious experimental research carried out by the authors and the abundance of experimental results interpreted in a rigorous scientific manner.
In order to improve the quality of the paper, I have just some minor suggestions.
- An image of the tribometer and some information about the data acquisition system (sensor type, acquisition rates, sensor precision) would be welcome.
- There is a wrong numbering of figures (Figures 10 to Figure 13), as Figures 10 and Figure 11 appear twice. The captions of these figures should be also checked. It was very difficult to follow the presentation from Section 3.3, as time as the reference to different figures was wrong. For this reason, Section 3.3 ought to be entirely amended.
- The Conclusion section should be enriched, as the authors presented a plenty of results, scarcely mentioned here.
Reviewer 3 Report
In this work, the authors prepared MoS2-WS2 composite films by magnetron sputtering and analysed their mechanical properties, and the tribological performance was tested under different atomic oxygen irradiation conditions. The content is expressed clearly and in proper English language, except for several typing errors that should be corrected (Line 30 in page 1, Line 100 in page 3, Line 210 in page 8, and Line 269 in page 10). The authors prepared the MoS2 film keeping the deposition time constant 20min to compare with the composite film but the thickness of MoS2 was not mentioned in the manuscript.
Reviewer 4 Report
Dear Authors,
Thanks for your great work on this study. I do have a couple questions about your work.
1- Line #21...It would be better if Autors mention the amount/percentage of the WS2.
2- Line #22. The authors mentioned that the addition of the WS2 improved the oxidation resistance. How much did the results improve the system? Please add numbers from your results.
3- Line #69: Why the system can not pump lower than 1.0x10-3 Pa. the system should go lower pressure (10-5 or 10-6 ) to eliminate the possible contaminations inside the air /chamber (even if you cleaned with Ar gas inside the chamber)?
4- Line #73- Why did you use very high pressure for the deposition?
5- Line #74... How did you determine the thickness of the sample? Have you used any measurement system to predict the thickness? Any reference point?
6- Figure 1: the target is facing toward to sample. Is the sample holder rotating? If rotate, how did you handle the flux distribution to the surface of the sample? If not rotate, you should erase the rotating motor part to eliminate the possible misunderstandings.
7- Line #180. Mentioned that during deposition, Oxygen atoms were replaced by the S atoms. If you decrease the pressure, you might minimize the Oxygen replacement during deposition. Have you tried lower pressure rather than 4.0 Pa?
8- In the Conclusion part, please mention the percentage change instead of just increased or improved.
Thanks
Round 2
Reviewer 1 Report
Thank you for making the necessary changes. As an overall recommendation, the paper is accepted.
Reviewer 4 Report
Dear Authors.
Thanks for the updates for my questions. You had a great job on your project.
Sincerely